# Molecular Responses of Vegetable, Ornamental Crops, and Model Plants to Salinity Stress

**DOI:** 10.3390/ijms24043190

**Published:** 2023-02-06

**Authors:** Stefania Toscano, Daniela Romano, Antonio Ferrante

**Affiliations:** 1Department of Science Veterinary, Università degli Studi di Messina, 98168 Messina, Italy; 2Department of Agriculture, Food and Environment, Università degli Studi di Catania, 95131 Catania, Italy; 3Department of Agricultural and Environmental Sciences—Production, Landscape, Agroenergy, Università degli Studi di Milano, 20133 Milan, Italy

**Keywords:** adaptive mechanisms, antioxidative metabolism, signal transduction, salinity-induced protein

## Abstract

Vegetable and ornamental plants represent a very wide group of heterogeneous plants, both herbaceous and woody, generally without relevant salinity-tolerant mechanisms. The cultivation conditions—almost all are irrigated crops—and characteristics of the products, which must not present visual damage linked to salt stress, determine the necessity for a deep investigation of the response of these crops to salinity stress. Tolerance mechanisms are linked to the capacity of a plant to compartmentalize ions, produce compatible solutes, synthesize specific proteins and metabolites, and induce transcriptional factors. The present review critically evaluates advantages and disadvantages to study the molecular control of salt tolerance mechanisms in vegetable and ornamental plants, with the aim of distinguishing tools for the rapid and effective screening of salt tolerance levels in different plants. This information can not only help in suitable germplasm selection, which is very useful in consideration of the high biodiversity expressed by vegetable and ornamental plants, but also drive the further breeding activities.

## 1. Introduction

Salinity is a major abiotic stresses that reduces crop productivity by hampering physiological processes in plants [1,2,3]. It affects approximately 19.5% of agricultural lands covering more than 830 million hectares [4]. Problems associated with salinity stress in the near future will expand due to global climate changes, in particular with the increase in temperature and reduction in water availability [5]. High salinity levels are a problem for vegetable and flower crops—which are irrigated crops—because soil salinity is often the result of incorrect irrigation practices that cause an increase in the concentration of salt in topsoil layers [6]. Vegetable and flower crops are often glycophytes and cannot grow in the presence of high levels of salt; a NaCl concentration over 100–200 mM often results in plant death [7]. The aesthetical appearance—and hence the cash value—of these crops [8,9] is strictly linked to the absence of salinity-induced damage, which is one of the principal morphological indicators of salt stress in plants [10]. Nonetheless, for many vegetable plants, more than twenty years after the review on the subject by Shannon and Grive [8], information on the effect of salinity stress is not sufficient. This lack of information is even more apparent for flower and ornamental crops.

The negative effects of salinization on plants include a reduction in growth trends due to the alteration of numerous biochemical and physiological processes [11]. Plant species perceive saline conditions and respond through a complex signaling network generated by ions, osmotic potential changes, and the biosynthesis of plant hormones or reactive oxygen species (ROS) [12]. These signals reach their respective receptors and determine the activation of physiological mechanisms that allow the plant to adapt to stress conditions. In the case of abiotic stresses, such as saline stress, three types of signal transduction have been identified, namely the ion signaling pathway, the osmolyte regulatory pathway, and the gene regulatory pathway [13]. Understanding each tolerance mechanism that occurs at the cellular level is essential for understanding how the entire plant organism reacts, which will contribute to improving the quality and productivity of different crops (Table 1) [3].

Therefore, this review aims to analyze the most recent results reported with respect to the molecular control of salt tolerance in vegetable and flower crops, in particular the osmotic regulation mechanism, salinity-induced proteins, reactive oxygen metabolism, the mechanism of signal transduction, and the development of salinity stress. This will provide useful indications to improve our knowledge of the mechanisms of salt tolerance in these species and assist in the development of future programs for the evaluation and selection of germplasm and genetic improvement.

## 2. Osmotic Regulation Mechanism

Salinity negatively affects plant physiology through ionic toxicity as well as osmotic and oxidative stress. Osmotic adaptation is essential to sustain cell turgor. Plants react to the osmotic stress caused by high salt levels mainly with osmotic adjustments (Figure 1).

Osmotic regulation is essential to maintain cell turgor and plant metabolic activity and, therefore, plant growth and productivity [45]. Plant under stress biosynthesizes a large number of osmoprotectants, such as proline (Pro), glycine betaine (GB), sugars, and sugar alcohols, facilitating the antioxidant mechanism and ionic homeostasis [46,47]. Pro is an amino acid and osmoprotectant, which acts as an important signaling molecule that accumulates in the cytosol of plants and functions in the stabilization and protection of membrane, protein enzymes, and various proteins. Pro also plays an important role during salinity stress by increasing the production of membrane proteins and ROS scavengers and maintaining cellular solute homeostasis. Many researchers have reported that under salinity, Pro improves water absorption and the antioxidant mechanism. Furthermore, it reduces the accumulation of toxic ions [46,48,49]. The Pro content that a plant accumulates to relieve osmotic stress can, therefore, be used as a physiological indicator of its ability to tolerate stress [50]. This substance exists in plant cells in the free state and has a low molecular weight, high water solubility, and no net charge in the physiological pH range [51].

Soluble sugars include mainly glucose, sucrose, and trehalose. These sugars can stabilize the cell membrane and protoplast [52]. They are osmolytes that stabilize the integrity of the cell membrane, which protects proteins from aggregation and denaturation. Soluble sugars can also be used as a physiological indicator of salt tolerance due to their osmoregulation function [1]. Sugar alcohols, such as inositol, sorbitol, and mannitol, also induce salinity tolerance by regulating cellular osmotic levels. These compounds increase the water potential of cells, allowing plants to uptake more water from the soil (Figure 1). In particular, they mitigate stress by promoting growth, scavenging ROS, maintaining cell turgor, and aiding in the sequestration of Na^+^ from the cytosol into the vacuole.

Betaines, such as glycine betaine (GB), proline betaine, hydroxyproline betaine, and pipecolate betaine, also act as osmoprotectants [48]. Of these, GB acts as a strong and compatible osmoprotectant in mitigating salinity stress. Extensive research has shown that GB maintains osmotic adaptation by regulating the Na^+^ to K^+^ ratio and accumulating in the cell; in particular, this reduces the toxic effects of the ions [53].

Further investigations are necessary to better understand the role of different osmoprotectants and the changes in gene expression in plants that occur as a response to this stress. Different genes are involved at the transcriptional level; it will be useful to discriminate if this gene expression is a direct result of the stress conditions or injury responses.

## 3. Reactive Oxygen Species Metabolism

Under abiotic stress conditions, photosynthetic activity is reduced, and the excitation energy of light becomes greater than that used or required by photosynthesis, resulting in an accumulation of ROS in the chloroplasts [54]. Abiotic stresses that induce the production of ROS hinder plant growth and lessen crop yield via reduction in chlorophyll contents and photosynthetic efficiency; the degree of this damage depends on the severity, frequency, and duration of the abiotic stress [55,56,57]. In optimal conditions, ROS are neutralized by intracellular antioxidants, whereas under salt stress conditions, the extreme accumulation of ROS produces oxidative stress and strongly disturbs the normal metabolism, causing protein destruction and nucleic acid mutation [58], and the overproduction of ROS in the plant induces lipid peroxidation in the cell membrane. The loss of membrane integrity impairs the physiological and biochemical processes in the plants. Under saline stress conditions, plants activate the production of malondialdehyde (MDA), which is the main product of membrane lipid peroxidation; its content represents the degree of cell membrane damage [1]. An increase in this compound is considered an oxidative stress indicator and is used as a tool to evaluate the tolerance of plants to salt conditions [59]. In salt stress conditions, plants increase the production of ROS, such as H_2_O_2_ (hydrogen peroxide), O_2_^−^ (superoxide), ^1^O_2_ (singlet oxygen), and ^∙^OH (hydroxyl radical). The overproduction of ROS then results in lipid peroxidation, protein degradation, and DNA mutation or damage [60]. To limit the oxidative damage caused by the excessive production of ROS, plants have implemented a complex antioxidant defense system which includes low-molecular-weight antioxidants (ascorbate, reduced glutathione, tocopherol, carotenoids, and flavonoids) and antioxidant enzymes, such as superoxide dismutase (SOD EC 1.15.1.1), peroxidase (POD EC 1.11.1.6), ascorbate peroxidase (APX EC 1.11.1.6), and catalase (CAT EC 1.11.1.6) [61].

The principal ROS-scavenging enzymes in plants include CAT, SOD, and POD. Catalase decomposes hydrogen peroxide into water and oxygen. In the same way, SOD converts superoxide radicals into oxygen and H_2_O_2_. POD and some other enzymes are involved in the degradation of H_2_O_2_ into innocuous products [62]. Ascorbate breakdown by APX involves the reduction of H_2_O_2_ to water [63].

SOD and catalase have been identified as the most effective enzymes in scavenging active oxygen species that cause oxidative stress. An alternative mode of H_2_O_2_ destruction is via peroxidases, which are found throughout the cell and have a much higher affinity for H_2_O_2_ than catalases [64]. Different studies have shown that salt stress treatment increases SOD activity [65]. Similarly, higher POD activity in the leaves of different plant species under salt stress conditions has also been found [66,67].

The principal site of ROS production under salinity stress conditions is the electron transport chain (ETC) in the chloroplast and mitochondria [68].

Controlling the production and action of ROS allows for a better understanding of the effects of various abiotic stresses on plants (Figure 2).

In various cultivated plants (rice, tomato, citrus, pea, and mustard), several studies have shown that the production of ROS is increased under salt stress conditions, and ROS-mediated membrane damage has been demonstrated as a major cause of cellular toxicity [69,70]. A significant increase in enzyme activities (SOD, POD, and CAT) was observed in the leaves of Asteraceae families of salt tolerant cv Wuxi [71]. Similarly, in sunflower leaves subjected to 200 mM NaCl, increases in enzyme activity and soluble protein content were observed [72]. In maize plants, Kaya et al. [73] observed that prolonged salinity reduced some physiological parameters (leaf relative water content and leaf water potential) and enhanced MDA and H_2_O_2_ concentration [73].

The overproduction of GSH and APX has been shown to improve oxidative stress tolerance, resulting in enhanced water stress in wheat [74]. One of the main causes of decreases in crop productivity is the production of ROS during abiotic stresses (salinity, water stress, and high and low temperatures). Hence, the regulation of ROS is a crucial process to avoid unwanted cellular cytotoxicity (DNA damage) and oxidative damage (Figure 2).

The metabolism of ROS under salt stress is complicated and includes mechanisms and interactions with other molecules and components that are yet to be identified. Thus, further studies are necessary to understand the process more clearly. Knowledge of ROS metabolism under salt stress will be helpful in mitigating salt-induced oxidative stress.

## 4. Mechanism of Signal Transduction and the Development of Salinity Stress

A high concentration of salts in soil or water induces several changes at the molecular level that allow crops to face the stress condition. Salinity is defined as the total amount of dissolved salts. The simplest method for salinity level determination is through electrical conductivity (EC). This index indirectly measures the salt concentration in soil or water. The most dangerous elements are sodium, chlorides, and sulphates. Plants have a very limited need for these elements; thus, they often accumulate in the soil near the roots and increase its salinity. However, high salinity can also be a result of excessive fertilizer application in cropping systems. The continuous application of mineral fertilizers to crops increases the salinity of the soil with time. Sodium (Na^+^) stress can be increased by irrigation water or infiltration of seawater along the coastal area. Seawater infiltration can reduce the number of crop species that can be grown as well as their productivity [75]. With climate change, the increase in global temperature in most Mediterranean areas will result in higher evapotranspiration rates in crops; this reduction in water availability can increase the salinity levels in soils and irrigation water. Crops can be exposed to salinity early in their life if they are planted in salty soil. In this condition, their germination can be inhibited by the salinity, and their signaling and activation systems can be slowed down [76]. On the contrary, crops can be exposed to a progressive salinity increase during development due to an increase in salts in the irrigation or underground water. In these cases, ion concentrations in the root zone gradually increase; plants can sense this salinity and activate signaling networks related to defense against excessive ion uptake and cellular accumulation.

The excessive uptake of ions can activate diverse protection mechanisms, and different plant species have varying defense strategies. Plants sense salinity through the cells of the roots that are in direct contact with salt ions [77].

Excessive salinity can induce osmotic stress in plants via the activation of salt-associated genes that function to mitigate the negative effects of Na^+^ accumulation. In this state, plants do not uptake water and reduce the leaf growth and cell elongation [78].

Na^+^ uptake can occur through ion transporters that have an affinity for monovalent cations, such as high-affinity potassium transporters (Figure 3). The accumulation of Na^+^ in the cytoplasm can inhibit the activities of enzymes, resulting in negative effects on cell metabolism. Decreasing the Na^+^ concentration in the cytoplasm can be achieved by storing this ion in the vacuole, a process carried out by Na^+^/H^+^ exchangers. If the Na^+^ concentration is also high in the vacuole, Na^+^/H^+^ exchangers can exclude the ion outside the cell (Figure 3). Several *Na^+^/H^+^ transporter (NHX)* genes have been identified in plants. These *NHX* genes are responsible for Na^+^/H^+^ uptake and they belong to the cation–proton antiporters 1 family. The translocation of the Na^+^ to the vacuole has the function of avoiding the inhibition of enzymatic reactions in the cytosol [79]. The ion homeostasis between Na^+^ and potassium (K^+^) in the cytoplasm is essential for maintaining vital physiological and biochemical processes. A low Na^+^ concentration in the cytoplasm can be achieved by the sequestration of the ion to the vacuole, while in the cytoplasm, K^+^ and calcium (Ca^2+^) increase in response to salinity [77].

Crops with high ionic uptake can adapt to salinity and maintain cell turgor by ionic compartmentalization in vacuoles and the accumulation of neutral solutes in the cytoplasm. Crops can avoid high ionic concentrations by direct salt extraction and controlling ionic uptake and translocation in growth zones. Crops can increase the volume of the areal part (succulence) as a dilution strategy [80]. Plants with excessive ionic uptake can experience negative effects due to ion accumulation in the cell wall and water deficits in the cytoplasm. The high accumulation of Na^+^ can impair metabolism and, in particular, photosynthesis.

Crops can adapt to high salinity, tolerate high ionic concentrations, and maintain turgor by compartmentalizing ions in the vacuoles and accumulating neutral solutes in the cytoplasm. Another strategy to avoid excessive ionic concentrations is the removal of salt from shoots by salt glands or phloem export or by the control of uptake and transport in growing areas.

Crops with reduced ionic uptake can adapt to salinity by avoiding drought stress. This strategy can be achieved by increasing the biosynthesis of organic solutes that maintain tissue turgor. In these crops, the cell wall acquires higher extensibility. Plant turgor can be preserved by increasing water permeability or leaf thickness to reduce transpiration. The use of water under high-salinity conditions is improved and crops show higher water use efficiency (WUE).

However, in crops that are sensitive to salinity stress, even low ion uptake can be damaged by high salinity as a consequence of a reduction in growth due to reduced turgor. Moreover, a reduction in CO_2_ fixation due to high stomatal resistance can limit plant development.

At the molecular level, specific genes have been shown to be associated with salinity and Na^+^ transport (Figure 4). Salt signaling is based on the plant’s sensitivity to salt. In *Arabidopsis thaliana* L., *Salt Overly Sensitive (SOS)* genes were isolated in mutants that were highly sensitive to salinity [81]. These mutants contributed to our understanding of how plants can defend themselves against excess salinity. *SOS* genes encode for specific proteins associated with Na^+^ transport. The *SOS1* gene is a Na^+^/H^+^ antiporter integrated in the plasmalemma that transports Na^+^ from the cytoplasm to the apoplast. The SOS2 and SOS3 proteins are located in the cytoplasm and associated with SOS1 function. Under high salt concentration, Ca^2+^ regulates the SOS3–SOS2 protein complexes, which block Na^+^ transporters to avoid its accumulation in the cell [82]. Functional studies have revealed that SOS1 is the key regulator of Na^+^ tolerance, while SOS2 and SOS3 play supporting roles. These Na^+^ transporters function in equilibrating the ion concentration in the cell [77]. Cytosolic Ca^2+^ is involved in the regulation of SOSs and tolerance to salinity stress conditions [83]. Ca^2+^ acts on SOS2 and SOS3, which block the uptake of Na^+^ [84]. Ca^2+^ seems to be regulated by the *reduced hyperosmolality-induced (Ca^2+^) increase1 (OSCA1)* gene. A functional analysis carried out by studying *osca1* mutants revealed that osmotic Ca^2+^ adjustment in the root and guard cells was disrupted, as was water flux regulation. Another type of channel for monovalent and bivalent cations is cyclic nucleotide-gated channels (CNGCs). In plants, 20 CNGCs have been found, and, in particular, CNGC18 is associated with Ca^2+^ activation [84].

Transcriptional studies performed under salinity conditions have reported that many transcription factors are activated by salinity, including *MYB2, MYC2, NAC, AREB/ABF, NAC (HD-ZIP),* and *DREB2* (Figure 4). These transcriptional changes regulate the expression of diverse genes, which leads to adaptation to salinity conditions. These genes are responsible for the activation of secondary genes that are associated with plant adaptation. From transcription to translation, the response can be very fast—from a few minutes or hours to days.

Transcriptional studies have elucidated the cluster of genes that is activated by salinity stress. In wheat, it was shown that the transcription factors induced by salinity included *MYB, NAC, bHLH, WRKY, bZIPs,* and *AP2*/ER (Figure 4). These genes regulate the expression of water channels (*aquaporins (AQ)*), *late-embryogenesis-abundant (LEA*) proteins, *dehydrins*, proline synthesis enzyme (*P5CS*), and proline oxidase (proline degradation enzyme). The activation of these genes increases the tolerance of wheat to salinity stress [85]. In addition, *SOS1, K^+^ transporters*, *glycerol-3-phosphate dehydrogenase (GPDH)*, Calcium ATPase, and *ABC transporters* were also found to be activated under salt stress. An analog transcriptomic study performed on *Solanum lycopersicum* L. and *S. chilense* (Dunal) Reiche at the vegetative stage showed differentially expressed salt stress genes. The most important cluster of genes was associated with antioxidant defense enzymatic mechanisms, including *SOD*, *CAT*, and *APX* [86]. In a transcriptional study performed to investigate the effect of borage extract under salinity, the transcription factors *DtRD29A* and *DtHB7* were highly expressed after 9 h under salinity conditions [87]. From a practical point of view, tolerance to salinity could be transiently activated by the application of calcium nitrate fertilizers. A similar effect can be obtained with potassium nitrate [88]. However, these two cations can act in different ways; Ca^2+^ plays a role in signaling and inhibiting Na^+^ accumulation, while K^+^ can reduce the Na^+^ concentration by competing for the same transport channels. There are also genes activated by salinity that are involved in membrane biosynthesis or repair, such as *phosphatidylinositol-specific phospholipase C (PI-PLC).* This gene activation is shared with different biosynthetic pathways.

Further investigations are required to identify the key genes that can induce tolerance to salinity stress in crops. Some salinity-associated genes have been found, and new undiscovered genes could be the master regulators of salinity tolerance in crops. The identification of these few genes could lead to the improved selection of new tolerant crops.

## 5. Salinity-Induced Proteins, Amino Acids, and Enzymes

Plants under abiotic stresses can accumulate a wide number of proteins that are shared among diverse stresses. Under salinity stress, plants can increase the accumulation of proteins as a storage form of nitrogen, which can be reused when the stress conditions are over. In particular, osmotins (26 kDa protein) are accumulated under salinity in diverse plant species [89]. Plants exposed to salinity also increase the accumulation of amino acids, the most important of which are proline, glutamine, and asparagine. Some of these are also associated with senescence, with the same function of translocating and storing nitrogen in the seeds, trunk, or branches.

Proline is one indicator of crop tolerance to abiotic stresses, in particular drought and salinity [90,91,92]. Proline plays a beneficial role in crops exposed to stress conditions. It is an excellent osmolyte, and its function can be summarized by its three major roles during abiotic stresses: it is a metal chelator, antioxidant compound, and a signaling molecule. The biosynthesis of proline from glutamic acid is carried out through the activities of the following enzymes: ∆’-pyrroline-5-carboxylate synthetase (P5CS, EC:2.7.2.11), which leads to ∆’-pyrroline-5-carboxylate (P5C) formation and the conversion to proline by ∆’-pyrroline-5-carboxylate reductase (P5CR, EC 1.5.1.2). An increase in biosynthetic proline enzymes in salinity conditions can help crops adapt to and tolerate stressful conditions.

In one study, overexpressing the Δ1-pyrroline-5-carboxylate synthetase gene in crops, such as soybean, induced an increase in tolerance to salinity, demonstrating the potential protection function of proline against salinity [93]. The endogenous proline concentration increased in plants exposed to salinity. *Catharanthus roseus* (L.) G. Don. ornamental plants under 80 mM NaCl stress also showed an increase in proline concentration [94]. Similar results were found in *Silvinia natans* (L.) All., which accumulated proline under high salinity stress [95]. On the basis of the observation of an increase in proline concentrations under salt stress, several studies have been performed to verify whether salinity tolerance can be achieved through exogenous applications for protecting crops. Exogenous treatments of proline can lead to a reduction in lipid peroxidation and programmed cell death under salinity stress [96]. In melon plants (*Cucumis melo* L.), the application of proline and potassium nitrate improved their tolerance to salinity (150 mM NaCl), avoiding yield reduction [97].

Salinity stress can activate several stress-related pathways, such as ethylene biosynthesis. Ethylene is a gaseous plant hormone that is biosynthesized via three enzymatic reaction steps. The first enzyme, S-adenosyl-methionine synthetase (S-AdoMet), converts the amino acid methionine into S-AdoMet, which is transformed into 1-Aminocyclopropane-1-carboxylate (ACC) by 1-aminocyclopropane-1-carboxylate synthase (ACC synthase, EC 4.4.1.14). ACC is directly converted into ethylene by an enzyme that oxidizes the ACC to form carbon dioxide and ethylene. This step is catalyzed by 1-aminocyclopropane-1-carboxylate oxidase (ACC oxidase, EC 1.14.17.4). Under salinity stress, the ethylene signaling network is activated with the involvement of its receptors, such as *ethylene receptor 1 (ETR1);* ethylene signaling messengers, such as *constitutive triple response 1 (CTR1);* and transcription factors, such as *ethylene insensitive (EINs)*, *ethylene insensitive lines (EILs)*, or *ethylene response factors (ERFs)*. Plants under salinity stress increase ethylene production, and the amount of ethylene produced is associated with salinity stress intensity. Experiments with Arabidopsis *etr1-1* mutants exhibiting reduced ethylene production showed a higher sensitivity to salinity. Therefore, treatments that alleviate salinity stress or increase salinity tolerance can be measured through the ethylene production.

Salinity tolerance or sensitivity in plants is also mediated by water balance, as previously described (Figure 5). At the biochemical level, it has been found that an increase in abscisic acid (ABA) in plants can help to counteract salinity stress. In tomato crops, an increase in ABA in rootstocks improved tomato tolerance. ABA is a plant hormone that is produced by carotenoid degradation (also known as the C40 indirect pathway). The key enzyme in this pathway is 9-cis-epoxycarotenoid dioxygenase (NCED, EC 1.13.11.51), which catalyzes the rate-limiting step in ABA biosynthesis. Transgenic tomato plants overexpressing the *SlNCED* gene exhibit a high ABA concentration, which helps the plants overcome transient stressful salinity conditions [98,99]. Several studies and review papers have suggested an important role of ABA in regulating crop tolerance to salinity [100]. Scientific evidence suggests that it can be used as a strategy in the selection of vegetable species or cultivars with higher ABA content or inducible salt-related ABA accumulation for growing vegetables under high-salinity conditions.

Another hormone involved in plant development is gibberellic acid (GA). This hormone has various different functions, but the most important is cell elongation. This effect can increase salinity tolerance in plants. One study performed on maize (*Zea mays* L.) demonstrated that treatment with GA_3_ (100 mg L^−1^) increased root and shoot biomass and PRO concentration and preserved membrane integrity in plants exposed to 100 mM NaCl [101]. Treatment with GA did not increase the activities of detoxification enzymes, indicating that the treated plants had lower stress. GA could act through the dilution effect on biomass development. Most investigations should be performed on the role of GA in protecting plants against salinity stress. Cytokinins are responsible for cell division and are leaf-yellowing inhibitors. In salinity stressed plants, the exogenous application of cytokinins can have direct or indirect effects. In marjoram (*Majorana hortensis* Moench) and spearmint (*Mentha spicata* L.) plants, the foliar application of cytokinins—in particular diphenylurea or kinetin—was able to counteract the negative effects of salinity on both plant growth and essential oil accumulation [102]. In rice (*Oryza sativa* L.), mutants that overaccumulated cytokinins showed a higher yield than wild-type stressed plants [103]. These results could be due to the effect of cytokinins on delaying senescence and increasing the concentrations of chlorophyll and the precursor of Pro. Brassinosteroids (BR) are involved in the plant hormones network (Figure 5) through their interaction with ethylene biosynthesis and can stimulate crop tolerance [104].

Metabolic profiles change under salinity stress, and further investigations should be performed to understand how biosynthesis or accumulation can be achieved using agronomic tools, such as plant growth regulators or biostimulants. Moreover, combined studies considering both transcriptomic and metabolomic data could help in the identification of the genes and metabolites that can be enhanced in new crops through specific breeding programs with the support of assisted selection of molecular markers.

## 6. Conclusions and Future Prospective

Salinity is strictly associated with irrigation and has a dramatic effect on crop productivity; hence, it plays an important role in food crises. This problem is particularly evident in vegetable and ornamental crops, which are irrigation crops that are required to be absent of salt damage due to the relevance of visual aspects in the quality of products. Moreover, many of them are glycophytic plants that are sensitive to salinity. The currently published data and scientific evidence suggest that crops have a predominant strategy that involves one or more pathways for salinity stress tolerance. Crops tolerant to salinity can have higher concentrations of antioxidant compounds or highly active enzymatic systems that regulate the biosynthesis of metabolites associated with osmotic adjustment.

A considerable number of questions and aspects need to be further investigated. Further information about the mechanisms of action of ROS, the role of osmoprotectants, and the mechanisms of signal transduction and salinity-induced substances are necessary. Omic approaches to these questions will be useful for improving our understanding of tolerance mechanisms and better orientating breeding activities by using new biotechnological solutions. This information can be exploited for selecting tolerant wild species that can be used as rootstock for grafting, as a source for tolerant traits, or for direct use in agricultural systems.

## Figures and Tables

**Figure 1 ijms-24-03190-f001:**
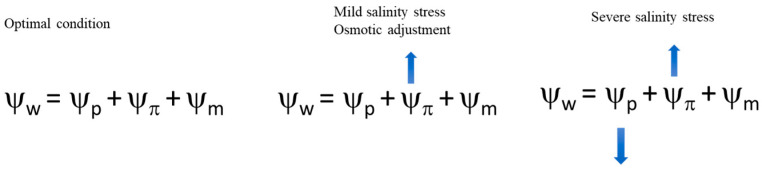
Water balance of crops under normal conditions and under progressive stress conditions (from left to right). The increase in salinity stress induces the accumulation of osmolytes that increase the ability of root cells to uptake water (the cell osmotic potential becomes more negative). The progression of salinity and its severity cannot be counteracted with only osmolytes; the plant does not uptake water and loses its turgor. Ψ_W_ = water potential; Ψ_p_ = pressure potential; Ψ_π_ = gravimetric potential; and Ψ_m =_ matric adsorption force.

**Figure 2 ijms-24-03190-f002:**
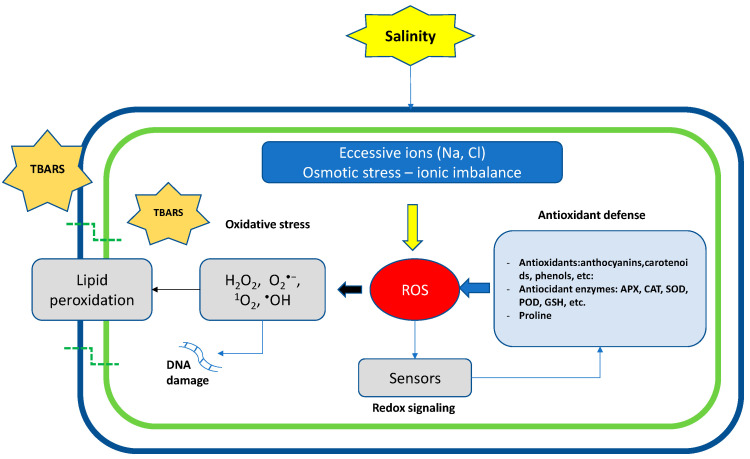
At the cellular level, salinity induces osmotic stress, ion imbalance, ROS accumulation, DNA damage, and lipid peroxidation. Crops activate redox signaling that leads to the induction of antioxidant defenses, such as the biosynthesis of antioxidants or the activation of antioxidant enzymatic systems.

**Figure 3 ijms-24-03190-f003:**
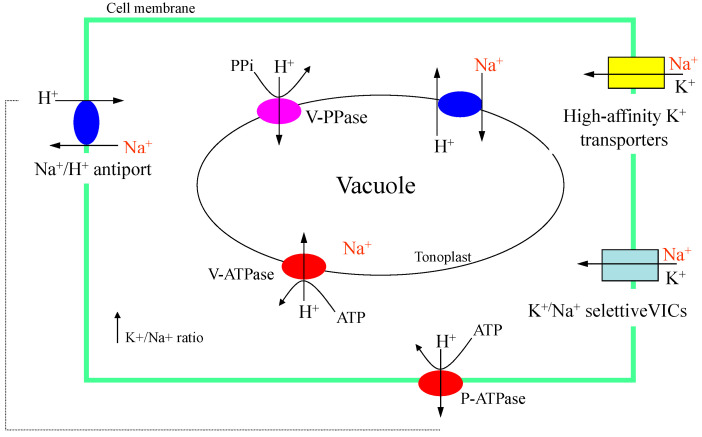
Possible Na^+^ uptake or extrusion in plant cells, including accumulation in the vacuole, for the protection of biochemical processes at the cytoplasm level.

**Figure 4 ijms-24-03190-f004:**
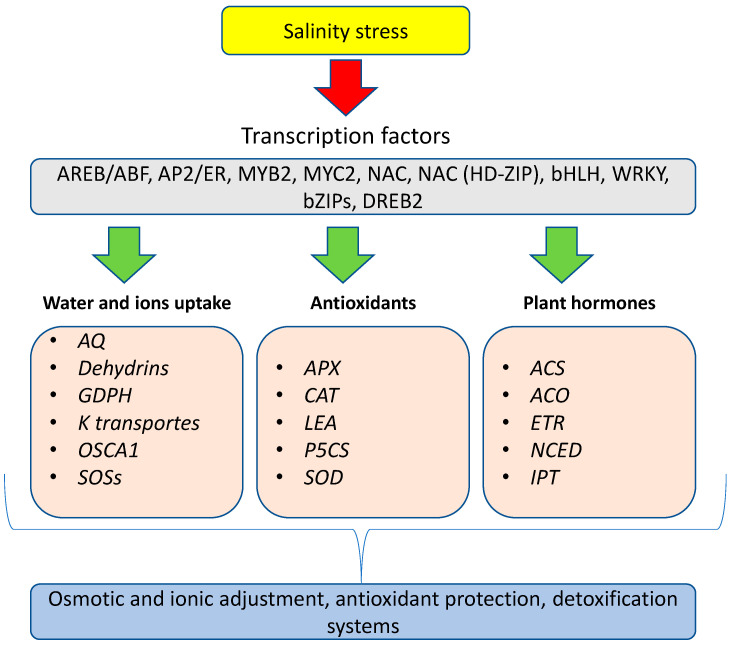
The first crop salinity responses at the molecular level involve the expression of transcription factors that activate the regulatory genes responsible for the activation of secondary genes. These clusters of genes are involved in water and ion uptake, the production of antioxidant enzymes, and plant hormones biosynthesis.

**Figure 5 ijms-24-03190-f005:**
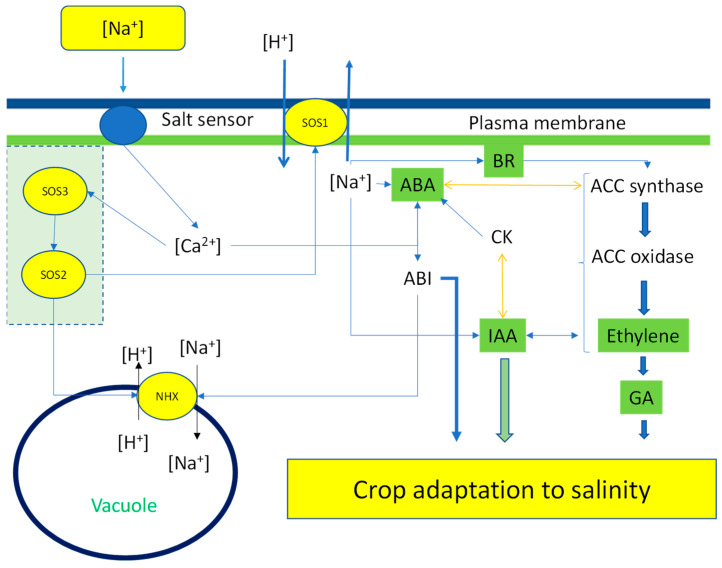
Under salinity stress due to high sodium concentration, the cell activates several protective systems that lead to adaptation. The sodium sensor promotes an increase in calcium in the cytoplasm that regulates the SOS1 and SOS3. These complexes are responsible for sodium accumulation in the vacuole and extrusion of sodium in the apoplast out of the cell. Sodium and calcium regulate the plant hormone network and interact with each other to allow for crop adaptation to high-salinity conditions.

**Table 1 ijms-24-03190-t001:** Main molecular effects of salt stress in different vegetable and ornamental crops.

Plant Species and Cultivar	Salt Stress Level	Effect of Salt Stress on Plants	References
*Aquilegia oxysepala* Trautv. & C.A.Mey., *A. parviflora* Ledeb., and *A. viridiflora* Pall.	5.0 ± 0.2 dS m^−1^ and 10.0 ± 0.2 dS m^−1^	Increase in MDA and proline; increased activity of SOD (5.0 dS m^−1^); *A. parviflora* POD increase; and *A. viridiflora* POD increase (10.0 dS m^−1^)	[14]
*Brassica oleracea* L.	80 mM	POX increase	[15]
*Brassica oleracea* L.	50, 100, 150, and 200 mM NaCl	Increase in CAT and POX activity, increase in proline	[16]
*Brassica rapa* L. subsp. *rapa* ‘Qiamagu’	50, 100, 150, and 200 mM NaCl	Increased activity of SOD (50, 100, 150, and 200 mM), POD (100, 150, and 200 mM), CAT (150 mM), and APX (200 mM); increase in MDA (100, 150, 200 mM)	[17]
*Brassica rapa* L. subsp. *rapa* ‘Wenzhoupancai’	50, 100, 150, and 200 mM NaCl	Increased activity in SOD and APX (200 mM), POD and CAT (100, 150, and 200 mM); increase in MDA (100, 150 mM)	[17]
*Calendula officinalis* L.	50–100 mM NaCl, 36 d	Increase in proline	[18]
*Calendula officinalis* L.	1, 5, and 9 dSm^−1^	Increase in MDA in leaves and roots, increase in proline in leaves (9 dS m^−1^), and increase in CAT activity; decrease in POD activity	[6]
*Capsicum annuum* L.	2000 and 4000 ppm NaCl	Increased activity of CAT and POX; increase in proline	[19]
*Capsicum annuum* L.	75 mM NaCl	Increase in SOD, POX, and CAT activity; increase in MDA	[20]
*Capsicum annuum* L. ‘Candy Apple’	35, 70, and 105 mM	Increase in APX and PPO activity	[21]
*Carthamus tinctorius* L.	50, 100, and 150 mM NaCl	Increased activity of CAT (50 mM), SOD (100 mM), and POD (50, 100, and 150 mM)	[22]
*Catharanthus roseus* (L.) G. Don	150 mM NaCl	MDA increase in vegetative and flowering stage; increase in CAT, GPX, and GR activity in vegetative and flowering stage	[23]
*Chrysanthemum* L. cvs. (‘Garden Beauty’, ‘Shanti’, ‘Red Stone’, ‘Basanti’, ‘Yellow Reflex’, ‘Ravi Kiran’, ‘Anmol’, ‘Mother Teresa’, ‘Sweta Singar’, and ‘Jaya’)	150 mM NaCl	Increase in proline	[24]
*Cornus florida* L. and *C. hongkongensis* subsp. *elegans* (Fang & Hsieh) Q.Y.Xiang	0.2%, 0.3%, 0.4%, and 0.45% salt solution	Increase in MDA, SOD activity (0.2%, 0.3%, 0.4%, and 0.45% salt solution) and proline (0.3%, 0.4%, and 0.45% salt solution)	[25]
*Cucumis melo* L.	30, 60, and 90 mM NaCl	Increase in proline, MDA, APX, CAT, SOD, and POD	[26]
*Cucumis sativus* L. (‘Green long’, ‘Marketmore’, ‘Summer green’, and ‘20252’)	NaCl 50 mM L^−1^	Increase in proline	[27]
*Dracaena braunii* Engl.	2.0 and 7.5 dS m^−1^	Increase in proline	[28]
*Fragaria ×ananassa* (Duchesne ex Weston) Duchesne ex Rozier ‘Gaviota’	50 mM	Increase in MDA	[29]
*Helianthus annuus* L. (ornamental sunflower)	150 mM NaCl	Increased activity of CAT and POD; increase of proline	[30]
*Lavandula multifida* L.	10–200 mM NaCl, 60 d	Increase in soluble sugars concentration	[31]
*Luffa acutangula* Roxb.	75 and 150 mM	Increase in proline	[32]
*Ocimum gratissimum* L. (African basil)	30, 60, 90, 120 mM	Increase in proline in leaves (120 mM) and root (90 and 120 mM)	[33]
*Oenanthe javanica* DC. ‘V11E0022’ and ‘V11E0135’	50 and 100 mM NaCl	Increase in leaves and roots of MDA and proline	[34]
*Pisum sativum* L. ‘L-888’ and ‘Round’	150 mM	‘L-888’ increase in CAT activity; ‘Round’ increase in proline and decrease in SOD activity	[35]
*Polianthes tuberosa* L.	50 and 100 mM NaCl	Increase in SOD, POD (100 mM), GR, and APX; increase in proline (100 mM)	[36]
*Portulaca oleracea* subsp. *oleracea* L., *P. grandiflora* Hook., *P. halimoides* L., and *P. oleracea* ‘Toucan Scarlet Shades’	100, 200, and 400 mM	Increase in proline in leaves and roots	[37]
*Rosa damascena* Mill. ‘Kashan’	4, 8, and 12 dS m^−1^	Increase in MDA (8 mM); increase in proline (8, 12 mM); and increase of CAT activity	[38]
*Solanum lycopersicum* L.	50 μM S-nitroso-N-acetyl penicillamine (SNAP) 200	Increased activity of APX, glutathione reductase (GR), peroxidase and rise in proline content	[39]
*Solanum lycopersicum* L.	100 mM	Mn-SOD, MDHAR, and GR decrease	[40]
*Solanum lycopersicum* L.	300 mM NaCl	Increase in proline	[41]
*Solanum lycopersicum* L.	150 mM NaCl	Increase in MDA content; increase in SOD and CAT activity	[42]
*Solanum lycopersicum* L. ‘Liaoyuanduoli’	150 mM NaCl	Increase in MDA and SOD, APX, GPX, GR, MDHAR, and DHAR activity	[43]
*Solanum lycopersicum* L. ‘Pusa Ruby’	150 mM NaCl	Increase in proline and MDA; increase activity of APX, MDHAR, DHAR, GR, SOD, CAT, GPX, and GST	[44]

## Data Availability

Not applicable.

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
