# Peer review of "Molecular Responses of Vegetable, Ornamental Crops, and Model Plants to Salinity Stress"

_ijms, 2023, doi:10.3390/ijms24043190_

Round 1
Reviewer 1 Report
In the submitted review manuscript Toscano et al., describe several aspects of plant response to salinity stress. This is an interesting and well-summarized review paper. However, I have found several points that should be improved as follows.
Three major concerns:
1, The whole paper is not well organized and needs extensive English editing.
2: All the major discussed mechanisms should be covered by figures or models to highlight the major points like ROS regulation, hormones, and genes regulated in response to salt stress.
3: The title does not cover the main points discussed in the main text, most part of this review paper is knowledge from crops and model plants Arabidopsis.
Minor concerns
1. Line 39: “the information is little or no sufficient” should be the information is little or not sufficient.
2. Line 42-45 “Plant species perceive saline stress and respond to it, thanks to complex signaling network generated by ions, the change of the osmotic differential, the biosynthesis of plant hormones or reactive oxygen species (ROS)” it does not Grammarly correct to me. Please restructure this sentence.
3. Line 53 flower crops plants, sound repetitive.
4. Table 1, the format of EC 5 in Salt stress level and the effect of salt stress on plants is not consistent with each other.
5. Space is missed in the Salt stress level of 50-100 mM NaCl, 36 d in the table.
6. Space is missed in Salt stress levels of 50, 100, and 150 mM NaCl in the table.
7. In table 1, mmol L−1 should change to mM throughout the manuscript.
8. Oenanthe javanica DC. ‘V11E0022’ and ‘V11E0135’ 50, 100, mm NaCl, highlighted should be mM.
9. In Figure 1, there should be a brief explanation and derivative of the formula. It is not easy for the reader to interpret it. What does the arrow stand for under the ѱp, in the third formula?
10. Line 74-76, Pro regulates the plant metabolism during stress by increasing membrane proteins, and ROS scavengers, and maintaining cellular solute homeostasis. This is quite obscure please rewrite it. Increase protein, kind, level, or activity?
11. Line 83-85, They serve as a strong osmolytes that stabilize the integrity of the cell membrane that protects proteins from aggregation and denaturation. Several grammar mistakes here, please rewrite it.
12. Line 92-93, Of these, GB acts as a strong and compatible osmoprotectors in mitigating salinity stress. Should be GB acts as a strong and compatible osmoprotector.
13. Line 97, subtitle 4. Reactive oxygen species metabolism should be subtitle 3. Reactive oxygen species metabolism.
14. Line 123, hydrogen peroxide (H2O2) should be H2O2.
15. Line 142-144, Kaya et al. [73] observed in maize plants that prolong salinity diminished some physiological parameters (leaf relative water content and leaf water potential) and enhanced MDA and H2O2 concentration [73]. Is highlighted and necessary?
16. Line 155, The most dangerous elements are sodium, chlorides, and sulfates. This claim does not seem right. As all of these three elements are essential elements for plant growth.
17. Line 159, Sodium (Na+) stress can increase through should be superscript (Na+).
18. Line 162-164, Under climate change the increase of global temperature in most Mediterranean areas can suffer of a higher evapotranspiration rate of crops and the reduction of water availability can increase the salinity levels in soils and irrigation water. Please rewrite this sentence. It isn't very clear.
19. Figure 2 is not fully explained, it should be more plausible to write a longer figure legend, more organelles should also be included such as the cell wall and nucleus. More importantly. ROS regulation should be included eighter combine with figure 2 or as a separate figure.
20. Line 182 and Line 184, Na+ should be superscript (Na+). Please check the whole draft.
21. Line 182 and Line 184, Ca++ should be superscript (Ca2+). Please check the whole draft.
22. Line 267, please give the full name of kDa.
Author Response
Dear reviewer,
The authors would like to thank you for your comments. The manuscript has been accordingly revised. Corrections and suggestions have been implemented in the current version of the manuscript. All the modifications are highlighted in yellow in the manuscript. We hereby provide a point-by-point answer.
The authors
In the submitted review manuscript Toscano et al., describe several aspects of plant response to salinity stress. This is an interesting and well-summarized review paper. However, I have found several points that should be improved as follows.
Three major concerns:
1: The whole paper is not well organized and needs extensive English editing.
A.A.: The manuscript has undergone by MPDI English pre-edit services.
2: All the major discussed mechanisms should be covered by figures or models to highlight the major points like ROS regulation, hormones, and genes regulated in response to salt stress.
A.A.: thank you for the suggestions for improving our manuscript. Models and figures have been added to support the descriptions of metabolites, plant hormones, and genes involved. A better description has been reported for the figure 2.
3: The title does not cover the main points discussed in the main text, most part of this review paper is knowledge from crops and model plants Arabidopsis.
A.A.: title has been changed considering the reviewer suggestion.
Minor concerns
Line 39: “the information is little or no sufficient” should be the information is little or not sufficient.
A.A.: the correction was done.
- Line 42-45 “Plant species perceive saline stress and respond to it, thanks to complex signaling network generated by ions, the change of the osmotic differential, the biosynthesis of plant hormones or reactive oxygen species (ROS)” it does not Grammarly correct to me. Please restructure this sentence.
A.A.: the sentence was rewrite
- Line 53 flower crops plants, sound repetitive.
A.A.: the sentence was rewrite
- Table 1, the format of EC 5 in Salt stress level and the effect of salt stress on plants is not consistent with each other.
A.A.: the table was corrected according your suggestion
- Space is missed in the Salt stress level of 50-100 mM NaCl, 36 d in the table.
- Space is missed in Salt stress levels of 50, 100, and 150 mM NaCl in the table.
- In table 1, mmol L−1 should change to mM throughout the manuscript.
- Oenanthe javanica DC. ‘V11E0022’ and ‘V11E0135’ 50, 100, mm NaCl, highlighted should be mM.
A.A.: All corrections you suggested are made in the table
- In Figure 1, there should be a brief explanation and derivative of the formula. It is not easy for the reader to interpret it. What does the arrow stand for under the ѱp, in the third formula?
A.A.: thanks for the suggestion; further information has been reported in the figure.
- Line 74-76, Pro regulates the plant metabolism during stress by increasing membrane proteins, and ROS scavengers, and maintaining cellular solute homeostasis. This is quite obscure please rewrite it. Increase protein, kind, level, or activity?
A.A. The sentence has been rewritten.
- Line 83-85, They serve as a strong osmolytes that stabilize the integrity of the cell membrane that protects proteins from aggregation and denaturation. Several grammar mistakes here, please rewrite it.
A.A. The sentence has been rewritten.
- Line 92-93, Of these, GB acts as a strong and compatible osmoprotectors in mitigating salinity stress. Should be GB acts as a strong and compatible osmoprotector.
A.A. Done.
- Line 97, subtitle 4. Reactive oxygen species metabolism should be subtitle 3. Reactive oxygen species metabolism.
A.A. Sorry for the mistake; the number has been corrected.
- Line 123, hydrogen peroxide (H2O2) should be H2O2.
A.A. The correction has been made.
- Line 142-144, Kaya et al. [73] observed in maize plants that prolong salinity diminished some physiological parameters (leaf relative water content and leaf water potential) and enhanced MDA and H2O2 concentration [73]. Is highlighted and necessary?
A.A. The sentence was modified.
- Line 155, The most dangerous elements are sodium, chlorides, and sulfates. This claim does not seem right. As all of these three elements are essential elements for plant growth.
A.A. The sentence was modified.
- Line 159, Sodium (Na+) stress can increase through should be superscript (Na+).
A.A: the sodium has been revised through the text.
- Line 162-164, Under climate change the increase of global temperature in most Mediterranean areas can suffer of a higher evapotranspiration rate of crops and the reduction of water availability can increase the salinity levels in soils and irrigation water. Please rewrite this sentence. It isn't very clear.
A.A. The sentence has been rewritten.
- Figure 2 is not fully explained, it should be more plausible to write a longer figure legend, more organelles should also be included such as the cell wall and nucleus. More importantly. ROS regulation should be included eighter combine with figure 2 or as a separate figure.
A.A. The legend of Figure 2 (now figure 1) was implemented. The figure for the ROS regulation was added (Figure 2: The salinity at cellular level induces osmotic stress, ions imbalance, ROS accumulation, DNA damage, and lipid peroxidation. Crops activate the redox signaling that leads to the antioxidant defense such as the biosynthesis of antioxidants or the activation of antioxidant enzymatic systems).
- Line 182 and Line 184, Na+ should be superscript (Na+). Please check the whole draft.
A.A: the Na+ has been revised through the text.
- Line 182 and Line 184, Ca++ should be superscript (Ca2+). Please check the whole draft.
A.A: the Ca2+ has been revised through the text.
- Line 267, please give the full name of kDa.
A.A.: Done
Reviewer 2 Report
The authors have reviewed the most recent advancements in the molecular response to salt stress in both vegetable and flower crop plants on several molecular mechanisms, including the osmotic regulation mechanism, the salinity-induced protein, the reactive oxygen metabolism, and the mechanism of signal transduction and development of salinity stress. I believe that the authors have provided sufficient background and explained well the current understanding of the topics that the authors have comprehensively reviewed. I have no major technical concerns about this manuscript but one major recommendation and several minor issues for the authors to consider if a revision is requested by the editor. I believe that these revisions would help significantly improve the overall presentation of this manuscript.
I would highly recommend that the authors should end each section by identifying the explicit areas that need to be explored further in the future studies in order to significantly advance the related areas. These areas should not be the same as the general descriptions given in Section 6.
Minor points:
Line 17: Should “vantage” be “advantage”?
Line 97, section 3 but not 4.
Lines 208-215: the format of these listed items should be changed to a description, instead of just a list.
The authors should be consistent with the use of gene names (italicized) and protein names (not italicized) throughout the entire manuscript.
Author Response
Dear reviewer,
The authors would like to thank you for your comments. The manuscript has been accordingly revised. Corrections and suggestions have been implemented in the current version of the manuscript. All the modifications are highlighted in yellow in the manuscript. We hereby provide a point-by-point answer.
The authors
The authors have reviewed the most recent advancements in the molecular response to salt stress in both vegetable and flower crop plants on several molecular mechanisms, including the osmotic regulation mechanism, the salinity-induced protein, the reactive oxygen metabolism, and the mechanism of signal transduction and development of salinity stress. I believe that the authors have provided sufficient background and explained well the current understanding of the topics that the authors have comprehensively reviewed. I have no major technical concerns about this manuscript but one major recommendation and several minor issues for the authors to consider if a revision is requested by the editor. I believe that these revisions would help significantly improve the overall presentation of this manuscript.
A.A.: Thanks for your positive comments.
I would highly recommend that the authors should end each section by identifying the explicit areas that need to be explored further in the future studies in order to significantly advance the related areas. These areas should not be the same as the general descriptions given in Section 6.
A.A.: Thanks for the suggestion; at the end of each section, the further studies and investigations have been reported.
Minor points:
Line 17: Should “vantage” be “advantage”?
A.A.: Sorry for the mistake; advantage has been added.
Line 97, section 3 but not 4.
A.A.: Sorry for the mistake; the number has been changed.
Lines 208-215: the format of these listed items should be changed to a description, instead of just a list.
A.A.: Thanks for the suggestion; the listed items were changed to a description.
The authors should be consistent with the use of gene names (italicized) and protein names (not italicized) throughout the entire manuscript.
A.A.: Thank you for the comments, the gene names have been revised and corrected in italics.
Round 2
Reviewer 1 Report
In the submitted review manuscript Toscano et al., describe several aspects of plant response to salinity stress. This is an interesting and well-summarized review paper. This revised draft addressed all my previous concerns. The whole draft was edited by professionals. New figures were also added to cover the major points this manuscript focuses on. The title and abstract are appropriate for the content of the text. I would like to suggest editor accept this paper.
1: Please use the same fonts for all figures.
2: Please check whether it should be “Excessive ions (Na+, Cl-) Osmotic stress – ionic imbalance” in figure 2?
Reviewer 2 Report
I appreciate very much the efforts that the authors have devoted to improving their manuscript. I have no more questions.